# Investigation on Nanocomposites of Polysulfone and Different Ratios of Graphene Oxide with Structural Defects Repaired by Cellulose Nanocrystals

**DOI:** 10.3390/polym15183821

**Published:** 2023-09-19

**Authors:** Yansong Yu, Yiwen Hu, Xiuduo Song, Jinyao Chen, Jian Kang, Ya Cao, Ming Xiang

**Affiliations:** 1State Key Laboratory of Polymer Materials Engineering, Polymer Research Institute, Sichuan University, Chengdu 610065, Chinachenjinyao@scu.edu.cn (J.C.); caoya@scu.edu.cn (Y.C.); xiangming@scu.edu.cn (M.X.); 2Key Laboratory of Combustion and Explosion Technology, Xi’an Modern Chemistry Research Institute, Xi’an 710065, China; huyiwenn123@163.com

**Keywords:** structural defect, graphene oxide, cellulose nanocrystal, polysulfone, nanocomposites

## Abstract

In this manuscript, nanofillers of graphene oxide (GO) and cellulose nanocrystal (CNC) with different weight ratios (G/C ratios), named GC 2:1, GC 4:1, GC 8:1, GC 16:1, and GC 32:1, were successfully prepared. Characterization methods such as Raman spectroscopy, X-ray photoelectron spectrometry (XPS), and thermogravimetric analysis (TGA) were performed. Additionally, the effects of these samples on the thermal stability, mechanical properties, and gas barrier properties of polysulfone (PSF) nanocomposites were investigated. A hydrophilic interaction took place between CNC and GO; as a consequence, CNCs were modified on the surface of GO, thus repairing the structural defects of GO. With the increase in G/C ratios, the repair effect of insufficient CNCs on the defects of GO decreased. The G/C ratio had a great influence on the improvement of mechanical properties, thermal stability, and gas barrier properties of nanocomposites. Compared with PSF/GC 2:1 and PSF/GC 32:1, the differences in the growth rates of tensile strength, elongation at break, and Young’s modulus were 30.0%, 39.4%, and 15.9%, respectively; the difference in T_d 3%_ was 7 °C; the difference in decline rate of O_2_ permeability was 40.0%.

## 1. Introduction

Polysulfone (PSF) has become a highly utilized membrane material due to its exceptional mechanical properties, excellent aging resistance, high heat resistance, outstanding processability, and numerous other superior features [1]. It has found wide application in numerous fields such as electronic appliances, machine industry, medical devices, and transportation [2,3]. Nevertheless, the poor resistance to fatigue, low interface energy, and susceptibility to fouling are major drawbacks of PSF membranes, which can seriously affect both the lifespan and the separation efficiency of the membranes [4]. In order to enhance the application properties of PSF membrane, it has recently become a popular trend to incorporate high-performance nanoparticles into the PSF membrane matrix [5,6].

Graphene, a kind of 2D nanomaterial composed of a hexagonal lattice of sp^2^ carbon atoms with a single-atom thickness [7], has gained prominence owing to its exceptional electrical, mechanical, thermal, and crystalline properties [8,9,10]. Therefore, it has been extensively investigated and applied in various fields, including energy devices, drug delivery, catalysts, polymer composites, sensors, and electronic materials [11,12,13,14]. As a result, there is an increasing consensus that graphene is an ideal nanofiller to reinforce polymer nanocomposites, owing to its abundant availability and exceptional functional performance [15,16].

Graphene oxide (GO) is the oxide form of graphene. It comprises oxidation regions and graphite zones. The edges of GO are composed of carbonyl and carboxyl groups, while the matrix contains hydroxyl and epoxides [17]. Therefore, due to its high hydrophilicity, graphene oxide exhibits excellent compatibility with polar polymers (PA, PMMA, PVA, PVC) [18]. As a result, GO is considered to be a suitable reinforcing nanofiller in polymer composites [19]. Furthermore, in order to enhance the compatibility between nonpolar polymers and GO, surface modification via the grafting of functional groups such as carboxyl and epoxide groups is necessary [8]. Functionalized GO (f-GO) demonstrates exceptional dispersibility and dissolubility in nonpolar polymers, e.g., polypropylene (PP), polystyrene (PS), and polyethene (PE) [20,21]. Therefore, PSF nanocomposites with great performance can be prepared by compositing PSF with GO [1,22].

There is cause for concern as the preparation process of graphite-based nanomaterials can lead to the formation of structural defects such as boundaries, grain macroscopic defects, Stone–Wales defects, and vacancies [23]. The presence of defective zones in 2D nanomaterials can impede the enhancement of desired performances of nanocomposites [23]. In the case of structural defects on the surface of graphene nanosheets, needless gas can flow directly into the composites instead of following a circuitous path [24]. Patching of structural defects and functionalizing graphite-based nanomaterials through heteroatom doping or grafting has been adopted as a mean of enhancing the performance of these materials [25,26,27].

Nanocellulose, a 1D nanomaterial with a high aspect ratio that is derived from sources such as cotton, wood, tunicates, and bacteria, has garnered widespread interest due to its renewable nature and exceptional performance attributes [28]. The classification of nanocelluloses includes cellulose nanofibril (CNF), bacterial cellulose (BC), and cellulose nanocrystal (CNC) [29]. Of these nanoparticles, CNCs are needle-shaped nanoparticles with high crystallinity, possessing exceptional strength and stiffness. These nanoparticles have a width of 5–20 nm and a length of 20–500 nm [30]. Furthermore, CNCs have the potential to significantly enhance the thermal stability and gas barrier properties of polymers, making them increasingly attractive for a variety of applications [31]. The hydroxyl groups present on the surface of CNCs govern their dispersion and provide sites for functionalization, allowing for a uniform dispersion of CNCs in various solvents [32].

Previous research has shown that strengthening 2D graphene-based nanoparticles with high-aspect-ratio 1D nanomaterials, such as CNCs, can enhance the interfacial energy between the particles [33]. CNC/graphene hybrids, as a promising class of advanced materials, have garnered significant attention due to their excellent physicochemical properties, biodegradability, and biocompatibility [34]. The hydrophilic interaction between CNC and rGO facilitates the nano-patching effect of CNC on rGO (reduced graphene oxide), resulting in the uniform dispersion of nanoparticles in their polyvinylidene chloride (PVDC) matrix. Similarly, in previous studies, the antibacterial properties of CNC/Ag/rGO hybrid PVA nanocomposites were enhanced [35]. Benefiting from the synergistic interaction, CNC/GO can serve as an effective nanofiller, enhancing the performance of solventless polymer nanocomposites [36]. Therefore, the synergistic effect of 1D and 2D nanoparticles in PSF matrices is expected to lead to substantial improvements in their performance [37]. However, the effect of the concentration ratios of GO to CNC (G/C ratios) on the repair effect has not been deeply studied.

To summarize, in this manuscript, the composite nanoparticles of GO with structural defects repaired by CNC with different G/C ratios were successfully prepared and characterized. The effect of these nanoparticles on thermal stability, mechanical properties, and gas barrier properties of PSF nanocomposites was also investigated. By adjusting the G/C ratio, it is possible to effectively enhance the defect repair efficiency of CNC toward GO.

## 2. Materials and Methods

### 2.1. Materials

Graphene oxide (GO) was obtained from Henan Angstron Graphene Technology Co, Ltd. (Luoyang, China). Polysulfone (Udel^®^ P-3500, Mn 22000) was obtained from Shanghai Solvay Co, Ltd. (Shanghai, China). N,N-Dimethylformamide (DMF) was obtained from Sinopharm Chemical Reagent Beijing Co, Ltd. (Beijing, China). Cellulose nanocrystals (CNC) were obtained from Huzhou ScienceK Technology Co, Ltd. (Huzhou, China). CNCs were prepared using the sulfuric acid hydrolysis method and obtained as a white powder through spray-drying. CNCs have a length ranging between 50 and 200 nm and a diameter between 5 and 20 nm. Structure diagrams of CNC and GO with structural defects are shown in Figure 1.

### 2.2. Sample Preparation

For the sake of preparing nanofillers of GO and CNC, GO/CNC weight ratios (G/C ratios) were varied to be 2:1, 4:1, 8:1, 16:1, and 32:1. The dispersion of GO and CNC (1 g) was added in 200 mL of DMF, and then subjected to ultrasonic dispersion at 60 °C for 30 min. The resulting mixture was filtered and dried under vacuum at 80 °C for 24 h. At last, GO/CNC nanofillers were successfully prepared, named GC 2:1, GC 4:1, GC 8:1, GC 16:1, and GC 32:1.

To prepare the PSF/GC composites, 200 mL DMF was first added to PSF (40 g) and stirred at 80 °C for 1 h. Thereafter, GC was added to the PSF dispersion at a loading of 0.5 wt.% (with respect to PSF) and stirred at 80 °C for 24 h. The obtained sample was then coated onto a glass plate. The PSF/GC composite membrane was obtained by peeling from the glass plate after drying at room temperature for 24 h. Using the same method, pure PSF, PSF/GO, and PSF/CNC membranes were also prepared.

### 2.3. Characterization Tests

#### 2.3.1. X-ray Photoelectron Spectra (XPS)

The XPS characteristics of the nanofillers, including GO, CNC, and GC with varying G/C ratios, were analyzed using the ESCALAB 250 photoelectron spectrometer (Thermo Fisher Scientific, Waltham, MA, USA). The instrument was operated with Al Kα radiation as the X-ray source (*hυ* = 1486.6 eV), with a power of 150 W and a pass energy of 30 eV for high-resolution scanning.

#### 2.3.2. Raman Spectra

Raman spectroscopy was performed on the nanofillers, including GO, CNC, and GC with varying G/C ratios, using a laser micro-Raman imaging spectrometer (DXRxi, Thermo Fisher Scientific, Waltham, MA, USA), measured using a linearly polarized 532 nm laser with 16 mW power. The samples had a thickness of approximately 0.1 mm [38].

#### 2.3.3. Thermogravimetric Analysis (TGA)

The TGA profiles of the nanofillers, including GO, CNC, and GC with varying G/C ratios, and nanocomposites, including PSF, PSF/GO, PSF/CNC, and PSF/GC with different G/C ratios, were determined using a TA Q5000IR thermo-analyzer (TA Instruments Inc., New Castle, DE, USA). The analysis was carried out from room temperature to 800 °C at a heating rate of 10 °C/min under a nitrogen atmosphere (with a flow rate of 100 mL/min).

#### 2.3.4. Testing Mechanical Properties

The mechanical properties of PSF nanocomposite membranes were determined using a CMT6104 microcomputer-controlled electronic universal testing machine (MTS) with a tensile speed of 10 mm/min under room temperature (25 °C). To this end, PSF membranes were cut into standard splines of dimensions 40 mm × 10 mm, and the average thickness of the splines was measured using the digital display spiral micrometer instrument. From the obtained typical stress–strain curves, the tensile strength, elongation at break, and Young’s modulus of the membranes were determined. For each PSF membrane sample, five specimens were tested, and the final data were taken as the average value.

#### 2.3.5. Oxygen Transmission Rate (OTR)

To determine the oxygen transmission rate (OTR), the OX-TRAN Model 2/21 (MOCON) was utilized in accordance with ASTM D3985 [39]. PSF membranes were cut into standard samples of dimensions 10.8 cm × 10.8 cm. Then, under 23 °C and 0% relative humidity conditions, the oxygen molecule transmission rate was measured by maintaining a certain pressure difference inside the container. On the basis of the collected data, numerical results for oxygen transmission rate per unit area could be obtained. The experiments were repeated five times to obtain reliable average values, and the corresponding range of error was reported.

## 3. Results and Discussion

### 3.1. Characterizations of Nanoparticles

#### 3.1.1. XPS Spectra

Figure 2 shows the peak separations of C1s for all samples, while Table 1 provides the corresponding relative atomic percentages. The XPS spectrum of raw GO (Figure 2a) exhibited a C1s peak that could be decomposed into five distinct curves, including characteristic peaks of carbon skeleton (C–C, 284.6 eV), carbon–oxygen single bond (C–O, 286.5 eV), carbon–oxygen double bond (C=O, 288.6 eV), and π–π* transition (291.5 eV) [40]. An important observation was the appearance of a 285.6 eV peak attributed to defects, revealing the presence of structural defects on GO [41]. In contrast, the XPS spectrum of CNC (Figure 2b) displayed a C1s peak decomposed into three curves: characteristic peaks of carbon skeleton (C–C, 284.6 eV), carbon–oxygen single bond (C–O, 286.5 eV), and O–C–O (288.1 eV).

When compared to GO, the XPS spectra of GC with different G/C ratios (Figure 2c–g) revealed the emergence of the characteristic peak of O–C–O and a significant reduction in the intensity of structural defects, providing evidence for hydrophilic interactions between CNC and GO, as well as the partial structural restoration of GO’s surface defects. Moreover, the order of intensity of characteristic peak of defects was as follows: GC 2:1 < GC 4:1 < GC 8:1 < GC 16:1 < GC 32:1. It can be inferred that, with the increase in G/C ratio, the amount of CNC was insufficient, resulting in the attenuation of the repair effect on defects of GO.

#### 3.1.2. Raman Spectra

Figure 3 shows the Raman spectra of GO and different GC samples. In all samples, two peaks were clearly visible: the D peak, corresponding to defects in the lattice of carbon atoms, located at approximately 1300 cm^−1^; the G peak, corresponding to the in-plane stretching vibration of carbon atoms bonded through sp^2^ hybrid orbitals, located at approximately 1600 cm^−1^ [42]. The defects of carbon atomic crystal can be characterized by the intensity ratio of D peak to G peak I_D_/I_G_ [43,44]. A larger ratio denotes a greater number of defects.

As can be seen from Figure 3, the order of I_D_/I_G_ of GC was as follows: GC 2:1 < GC 4:1 < GC 8:1 < GC 16:1 < GC 32:1 < GO. This is conclusive evidence that CNC repaired the structural defects on the surface of GO, thus reducing the I_D_/I_G_. 

#### 3.1.3. Thermogravimetric Analysis (TGA)

The thermogravimetric curves of GO, CNC, and GC with different G/C ratios are shown in Figure 4. It can be seen that, for CNC, the dehydration and decomposition of the cellulose chain took place at around 160 °C, followed by the pyrolysis of labile oxygen-containing functional groups that took place at around 280 °C. It can be proven that the thermal stability of CNC was much weaker than that of GO. Nevertheless, the thermal stability of GC was vastly improved when compared with CNC, even reaching a level close to that of GO. No distinct decomposition peaks were observed at 160 °C and 280 °C, as the presence of GO prevented the removal of functional groups from CNC at high temperatures. This is indicative of the presence of hydrophilic interactions between CNC and GO, which enhanced the thermal stability of GC nanoparticles. In addition, relatively speaking, the thermal stability of GC 32:1 was the best, while the thermal stability of GC 2:1 was the worst, which also proves that G/C ratios had a significant impact on the interaction between GO and CNC.

### 3.2. Properties of Nanocomposites

#### 3.2.1. Mechanical Properties

Typical stress–strain curves of various nanocomposites are shown in Figure 5, and the tensile strength, elongation at break, and Young’s modulus of different nanocomposites are shown in Figure 6 and presented in Table 2.

Apparently, the addition of nanoparticles led to an enhancement of the mechanical properties of PSF nanocomposites. Nonetheless, it should be noted that the elongation at break of pure PSF was higher than that of PSF/GO, owing to the uneven distribution of GO within the PSF matrix. Furthermore, it is noteworthy that the mechanical properties of PSF/GC showed the most significant improvement compared to the other nanocomposites. This can be attributed to the CNC repairing the structural defects on the surface of GO, resulting in a synergistic effect and enhanced compatibility between the nanoparticles and the PSF matrix. As a result, the mechanical properties of the nanocomposites were significantly enhanced. In addition, the order of mechanical properties of nanocomposites was as follows: PSF/GC 2:1 > PSF/GC 4:1 > PSF/GC 8:1 > PSF/GC 16:1 > PSF/GC 32:1. Compared with PSF/GC 2:1 and PSF/GC 32:1, the differences in growth rates of tensile strength, elongation at break, and Young’s modulus were 30.0%, 39.4%, and 15.9%, respectively. This proves that the G/C ratio had a great influence on the performance improvement of nanocomposites. A lower G/C ratio indicated a lower number of defects of GO and a better improvement effect on the mechanical properties of nanocomposites.

#### 3.2.2. Thermal Stability

The thermogravimetric curves of various nanocomposites with different G/C ratios are shown in Figure 7. The thermal stability of nanocomposites could be characterized by the temperature at which the mass loss was 3% T_d 3%_. Figure 8 and Table 3 show the variation of T_d 3%_ with the concentration of nanofillers.

Figure 8 and Table 4 demonstrate that the addition of nanoparticles led to an increase in T_d 3%_ of all nanocomposites. Notably, the increase in T_d 3%_ of PSF/GC was the most significant, which could be attributed to the repair effect and synergistic effect brought by GO and CNC. The order of T_d 3%_ was as follows: PSF/GC 2:1 > PSF/GC 4:1 > PSF/GC 8:1 > PSF/GC 16:1 > PSF/GC 32:1. Compared with PSF/GC 2:1 and PSF/GC 32:1, the difference in T_d 3%_ was 7 °C. This proves that the G/C ratio had a great influence on the improvement of thermal stability of nanocomposites.

#### 3.2.3. Gas Barrier Properties

Figure 9 and Table 4 show the O_2_ permeability of various nanocomposites.

From Figure 9, it can be seen that the O_2_ permeabilities of nanocomposites all deceased with the addition of nanoparticles. It can be inferred that the addition of CNCs to the nanocomposites played a vital role in repairing the structural defects on the surface of GO, thereby preventing the direct penetration of gas molecules through the defects. Instead, the gas molecules were forced to pass through the nanocomposites along the bending path, which effectively reduced the O_2_ permeability of the nanocomposites. Figure 10 figuratively shows the structure diagrams of GC with different G/C ratios. Similar to the above conclusion, the order of O_2_ permeability was as follows: PSF/GC 2:1 > PSF/GC 4:1 > PSF/GC 8:1 > PSF/GC 16:1 > PSF/GC 32:1. Compared with PSF/GC 2:1 and PSF/GC 32:1, the difference in decline rate of O_2_ permeability was 40.0%, which proves that the G/C ratio had a great influence on improving the gas barrier properties of nanocomposites.

Figure 10 shows the structure diagrams of GC with a low G/C ratio (2:1) and high G/C ratio (32:1).

The difference in gas barrier performance of the PSF nanocomposites with different G/C ratios can be explained from several aspects. Firstly, as an inorganic nanofiller dispersed in the PSF matrix, GC blocks the diffusion of oxygen molecules in the matrix, forcing them to bypass and take a more tortuous path, thereby prolonging the time for oxygen molecules to pass through the matrix and reducing oxygen permeability. However, for GC with a high G/C ratio, the low content of CNC results in a low defect repair efficiency, and the unrepaired structural defects on the GO surface allow oxygen molecules to penetrate, greatly reducing the barrier effect of GO on oxygen molecules. In contrast, for GC with a low G/C ratio, the structural defects on the GO surface are effectively repaired, preventing oxygen molecules from penetrating through the defects, and forcing them to continue to traverse the composite material along a tortuous path, thereby further improving the oxygen barrier performance of the PSF composite membrane.

## 4. Conclusions

In our study, the structural defects of GO were successfully repaired through the incorporation of CNC. Nanoparticles with different G/C ratios (GC 2:1, GC 4:1, GC 8:1, GC 16:1, GC 32:1) were successfully prepared and characterized. This study also examined the effects of these modifications on the physical properties, thermal stability, and gas barrier properties of PSF nanocomposites. The main conclusions are summarized below.

Through characterization tests, it was observed that there was indeed an interaction between graphene oxide and cellulose nanocrystals, which could be attributed to the interaction between oxygen-containing functional groups on their surfaces. The hydrophilic interaction between GO and CNC resulted in the surface modification of CNCs on the GO substrate. These modified CNCs played an essential role in repairing the structural defects present on the surface of GO. The order of I_D_/I_G_ of GC was as follows: GC 2:1 < GC 4:1 < GC 8:1 < GC 16:1 < GC 32:1. With the increase in G/C ratio, the repair effect of insufficient CNC on the defects of GO decreased.

The order of application performance was as follows: PSF/GC 2:1 > PSF/GC 4:1 > PSF/GC 8:1 > PSF/GC 16:1 > PSF/GC 32:1. Compared with PSF/GC 2:1 and PSF/GC 32:1, the differences in growth rates of tensile strength, elongation at break, and Young’s modulus were 30.0%, 39.4%, and 15.9%, respectively; the difference in T_d 3%_ was 7 °C; the difference in decline rate of O_2_ permeability was 40.0%. The G/C ratio had a great influence on improving the mechanical properties, thermal stability, and gas barrier properties of nanocomposites. By adjusting the G/C ratio, it was possible to effectively enhance the defect repair efficiency of CNC toward GO, thereby more effectively improving the performance of the PSF matrix.

## Figures and Tables

**Figure 1 polymers-15-03821-f001:**
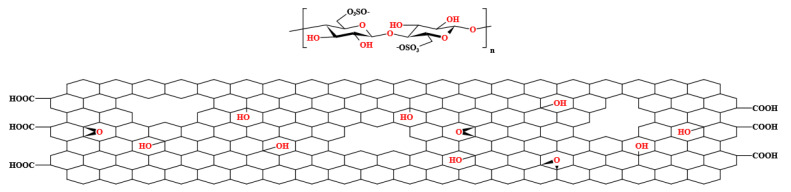
Structure diagrams of CNC and GO with structural defects.

**Figure 2 polymers-15-03821-f002:**
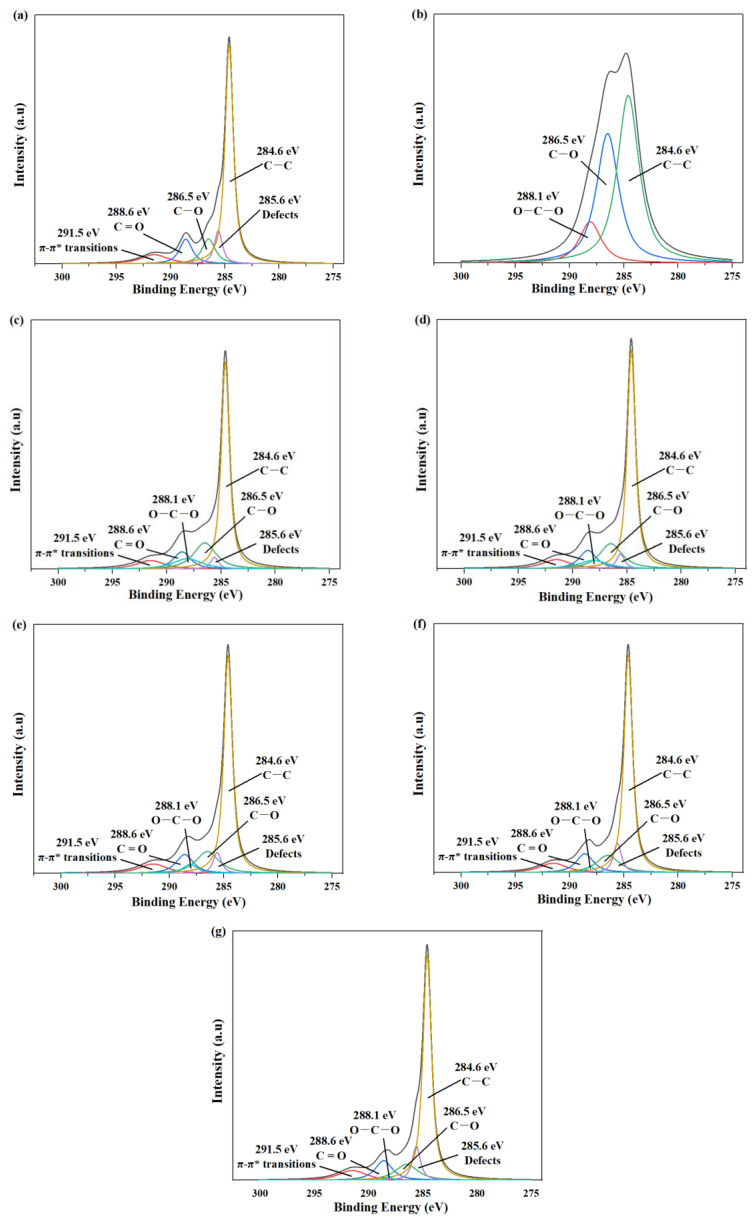
The peak separations of C1s: (**a**) GO; (**b**) CNC; (**c**) GC 2:1; (**d**) GC 4:1; (**e**) GC 8:1; (**f**) GC 16:1; (**g**) GC 32:1.

**Figure 3 polymers-15-03821-f003:**
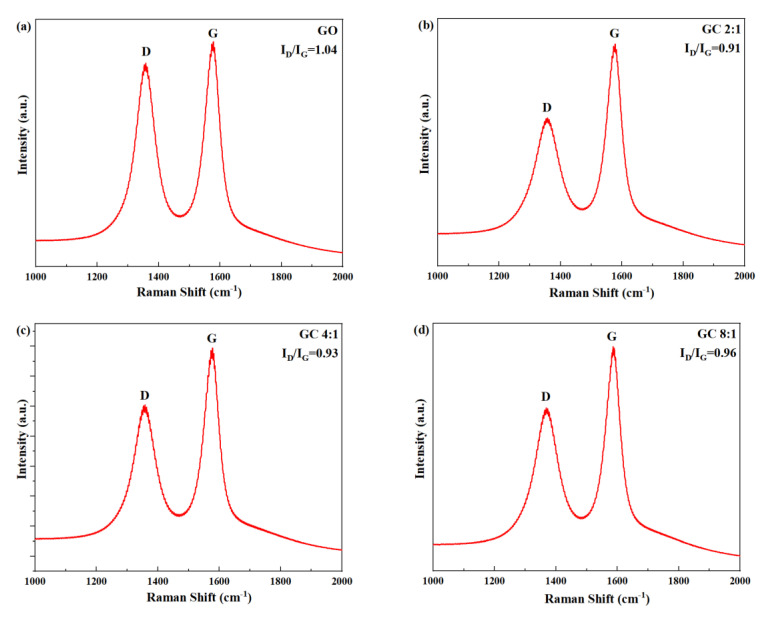
Raman spectra of GO and GC with different G/C ratios: (**a**) GO; (**b**) GC 2:1; (**c**) GC 4:1; (**d**) GC 8:1; (**e**) GC 16:1; (**f**) GC 32:1.

**Figure 4 polymers-15-03821-f004:**
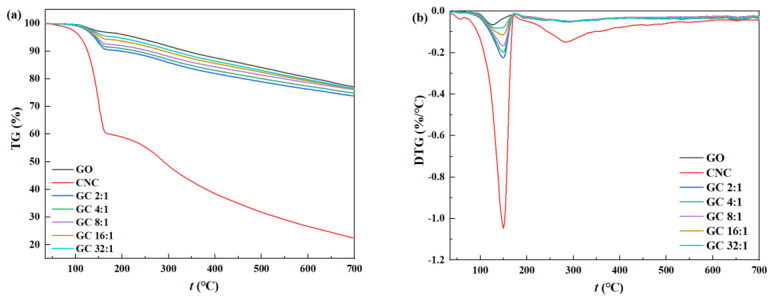
The thermogravimetric curves of GO, CNC, and GC with different G/C ratios: (**a**) TGA; (**b**) DTGA.

**Figure 5 polymers-15-03821-f005:**
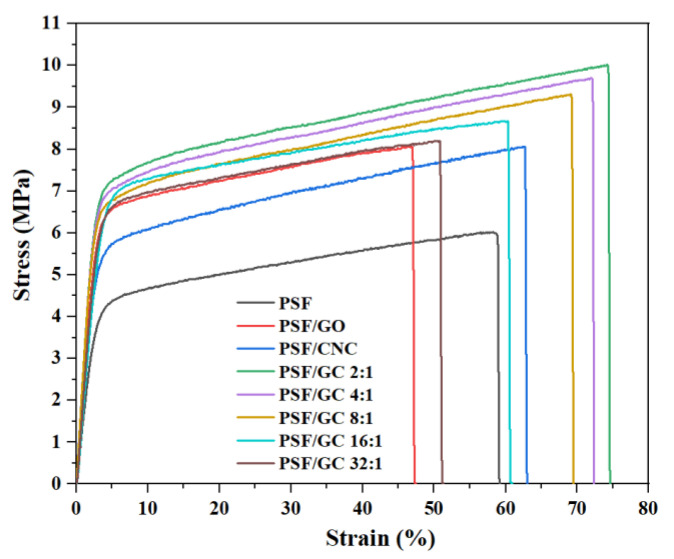
Typical stress–strain curves of various PSF nanocomposites with different G/C ratios.

**Figure 6 polymers-15-03821-f006:**
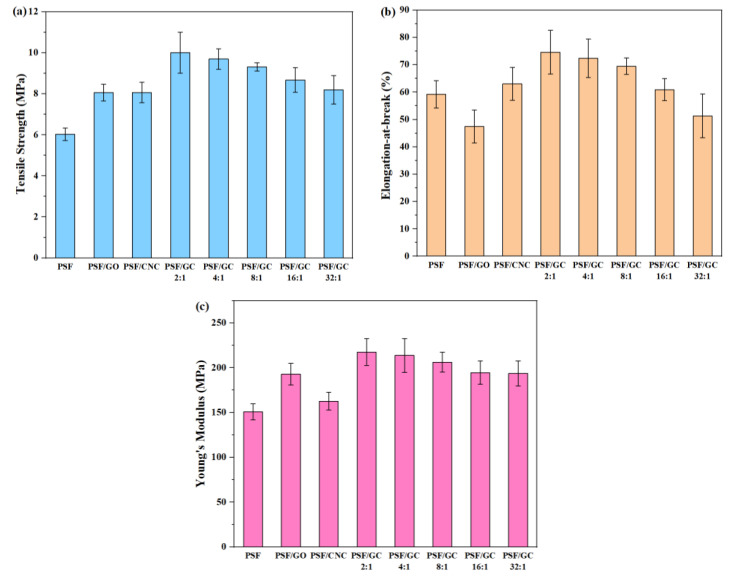
Mechanical properties of various PSF nanocomposites with different G/C ratios: (**a**) Tensile strength; (**b**) elongation at break; (**c**) Young’s modulus.

**Figure 7 polymers-15-03821-f007:**
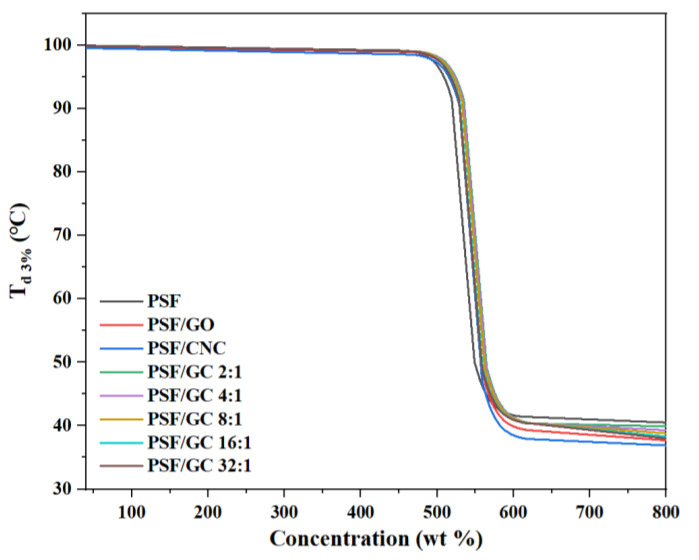
Thermogravimetric curves of various PSF nanocomposites with different G/C ratios.

**Figure 8 polymers-15-03821-f008:**
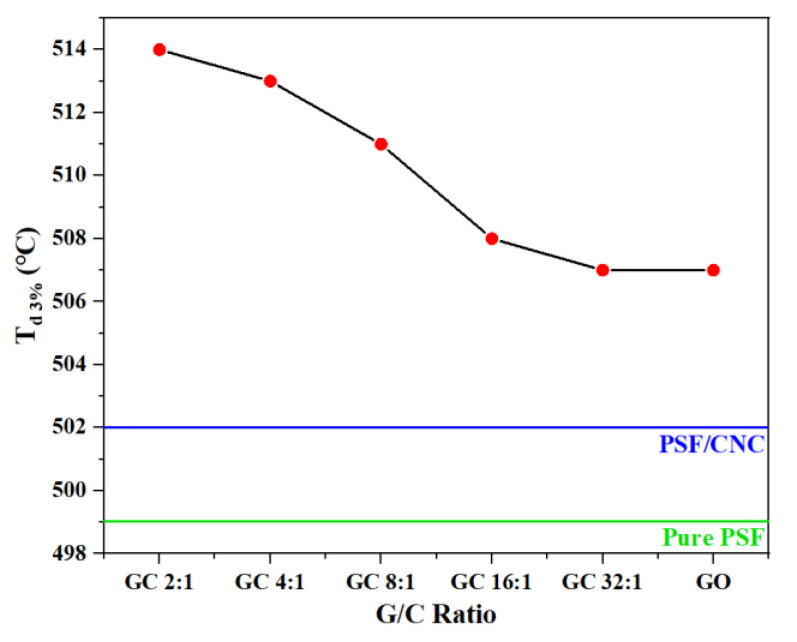
The temperature at which the mass loss was 3% T_d 3%_ as a function of G/C ratio.

**Figure 9 polymers-15-03821-f009:**
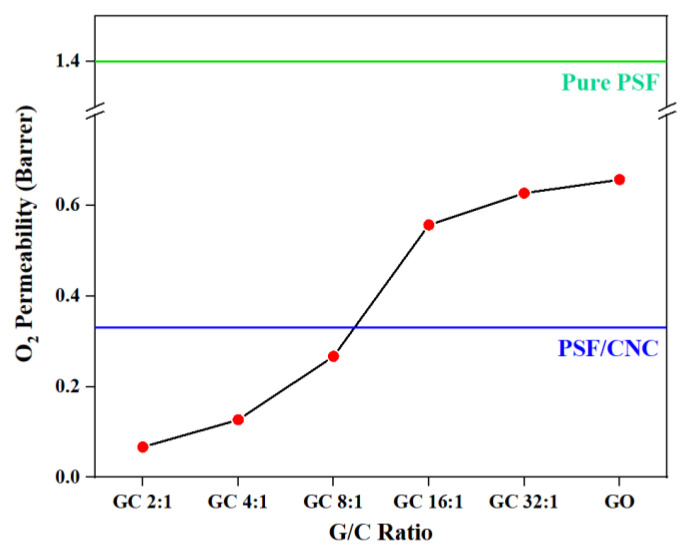
O_2_ permeability of various PSF nanocomposites with different G/C ratios.

**Figure 10 polymers-15-03821-f010:**
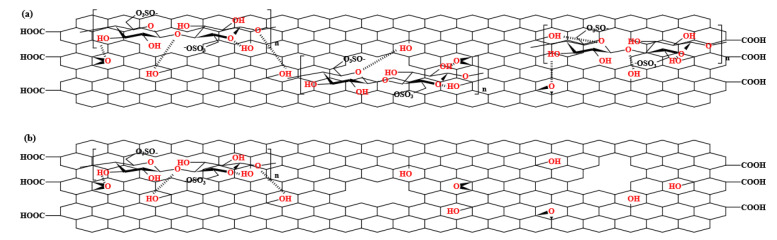
Structure diagrams of (**a**) GC 2:1, and (**b**) GC 32:1.

**Table 1 polymers-15-03821-t001:** The relative atomic percentages for GO, CNC and GC with different G/C ratios.

Sample Name	C–C (%) 284.6 eV	C–O (%) 286.5 eV	C=O (%) 288.6 eV	π–π* (%) 291.5 eV	Defects 285.6 eV	O–C–O (%) 288.1 eV
GO	60.5	11.2	10.7	8.5	9.1	0
CNC	49.0	38.9	0	0	0	12.1
GC 2:1	52.6	18.2	9.2	7.4	2.9	9.5
GC 4:1	54.5	17.1	9.6	7.7	3.6	7.4
GC 8:1	57.2	15.3	10.1	8.1	5.3	4.0
GC 16:1	59.3	12.6	10.5	8.4	7.7	1.5
GC 32:1	60.2	11.4	10.6	8.5	8.7	0.6

**Table 2 polymers-15-03821-t002:** Tensile strength, elongation at break, and Young’s modulus of various PSF nanocomposites with different G/C ratios.

Sample Name	Tensile Strength (MPa)	Elongation at Break (%)	Young’s Modulus (MPa)
PSF	6.0 ± 0.3	59.2 ± 5.2	150.7 ± 9.5
PSF/GO	8.1 ± 0.4	47.4 ± 6.7	192.9 ± 12.5
PSF/CNC	8.1 ± 0.6	63.0 ± 6.4	162.6 ± 10.3
PSF/GC 2:1	10.0 ± 1.0	74.6 ± 8.3	217.6 ± 15.8
PSF/GC 4:1	9.7 ± 0.6	72.4 ± 7.4	213.8 ± 19.1
PSF/GC 8:1	9.3 ± 0.3	69.5 ± 3.8	206.3 ± 11.4
PSF/GC 16:1	8.7 ± 0.6	60.9 ± 4.2	194.6 ± 13.6
PSF/GC 32:1	8.2 ± 0.8	51.3 ± 8.4	193.7 ± 14.5

**Table 3 polymers-15-03821-t003:** T_d 3%_ of various PSF nanocomposites with different G/C ratios.

Sample Name	T_d 3%_ (°C)	Changes (°C)
PSF	499	-
PSF/GO	507	+8
PSF/CNC	502	+3
PSF/GC 2:1	514	+15
PSF/GC 4:1	513	+14
PSF/GC 8:1	511	+12
PSF/GC 16:1	508	+9
PSF/GC 32:1	507	+8

**Table 4 polymers-15-03821-t004:** O_2_ permeability of various PSF nanocomposites with different G/C ratios.

Sample Name	O_2_ Permeability (Barrier)	Changes (%)
PSF	1.40	-
PSF/GO	0.66	−53.1
PSF/CNC	0.33	−76.2
PSF/GC 2:1	0.07	−95.2
PSF/GC 4:1	0.13	−90.9
PSF/GC 8:1	0.27	−80.9
PSF/GC 16:1	0.56	−60.2
PSF/GC 32:1	0.63	−55.2

## Data Availability

The data are not publicly available due to privacy.

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
