# Peer review of "Investigation on Nanocomposites of Polysulfone and Different Ratios of Graphene Oxide with Structural Defects Repaired by Cellulose Nanocrystals"

_polymers, 2023, doi:10.3390/polym15183821_

Round 1

Reviewer 1 Report (Previous Reviewer 3)

The manuscript is revised along the suggested lines so can be accepted. 

Author Response

Reviewer 2 Report (Previous Reviewer 2)

Accept in present form.

Minor editing required

Author Response

Reviewer 3 Report (New Reviewer)

The work presents a comprehensive study on the incorporation of cellulose nanocrystals to repair structural defects in graphene oxide within a polysulfone matrix. The authors use various characterization techniques to assess the nanocomposite's properties. They establish a clear relationship between the G/C ratio, defect repair efficiency, and resulting material properties. The study addresses important aspects of materials science, including nanocomposite synthesis, characterization, and property evaluation. The use of different ratios allows for a thorough analysis of the relationship between nanofiller content and resulting properties. This work could potentially be published after considering the following point in detail or properly: 

  1. Characterization Techniques: The study relies on Raman spectra, X-ray photoelectron spectroscopy (XPS), and thermogravimetric analysis (TGA) for characterization. While these techniques are commonly used, they might not offer a complete understanding of the nanocomposite's structure and properties. Additional techniques, such as transmission electron microscopy (TEM) or atomic force microscopy (AFM), could provide more detailed information on the dispersion and interactions between nanofillers and the polymer matrix.
  2. Sample Preparation Reproducibility: The description of the sample preparation methods lacks certain crucial details. Factors like sonication parameters, dispersion time, and temperature during sample drying can significantly influence the resulting nanocomposite's properties. Inadequate control and reporting of these parameters can lead to inconsistencies and difficulties in reproducing the results.
  3. Data Quantification and Statistical Analysis: The presented data appears to be qualitative in nature, and quantitative analysis might be limited. To strengthen the conclusions, statistical analysis and multiple replicates should be incorporated wherever applicable. This would enhance the robustness of the results and provide a better understanding of the variations observed.
  4. Limited Insight into Mechanisms: While the study demonstrates the repair effect of CNC on GO defects and its influence on material properties, it lacks an in-depth exploration of the underlying mechanisms driving these changes. Providing insights into the molecular interactions between CNC, GO, and the polymer matrix would add more depth to the understanding of the repair process.
  5. Long-Term Stability: The work primarily focuses on immediate improvements in properties resulting from the repair process. However, the long-term stability and durability of the nanocomposite materials should also be considered, as interactions between nanofillers and the matrix, as well as potential changes over time, could affect material performance.
  6. Comparison with Existing Literature: While the study highlights the improvements achieved with different G/C ratios, it might benefit from a more extensive comparison with existing literature on similar nanocomposite systems. This would help contextualize the findings and determine the novelty and significance of the work.
  7. Biocompatibility and Environmental Impact: If the application of these nanocomposites extends to fields involving human contact or the environment, it's essential to consider the biocompatibility of the materials and their potential environmental impact. These aspects are not addressed in the current abstract.
  8. Improve the image resolution. Give a detailed caption for each figure and table.
  9. Conclusions must be restructured by considering the previous points.
  10. Figures 1 and 2 do not make sense to illustrate the distribution of functional groups on the graphene surface as this fact is not supported with the current results. 

To enhance the English writing, strive for clear and precise language with varied sentence structures. Ensure correct grammar and punctuation, incorporate transition words for smooth flow, and opt for active voice. Select precise vocabulary, maintain consistent terminology, and proofread meticulously for errors. These improvements will bolster readability and communication effectiveness.

Author Response

Reviewer 4 Report (New Reviewer)

The document shows how a mixed nanofiller of GO and CNC can reinforce PSF films. The idea is innovative and may open room for new materials. The manuscript is well-written and logical. There are just a few points that the authors must rework. I hope my comments are helpful.

**Title:

-The title gives much attention to the GO/CNC nanofiller but not much to the matrix. If the authors agree, I would like to suggest the following title:

“Investigation on Nanocomposites of Polysulfone and Different Ratios of Graphene Oxide with Structural Defects Repaired by Cellulose Nanocrystals.”

**Abstract:

-Please define PSF (line 15). Please remember to define the abbreviations cited in the abstract.

**Keywords: OK

1. Introduction

-The current flow is:

PSF advantages and limitations > Chemical properties and advantages of graphene > Chemical properties of GO > Defects formed on 2D-nanomaterials > Chemical properties and uses of CNCs > Advantages and challenges of mixing 1D and 2D nanomaterials > Objectives of this work.

The flow is logical and very good. There is no need to change it.

-Please define rGO. 

-It seems that the last paragraph has writing errors. Please recheck it.

2. Experimental

2.1. Materials:

-Please write a sole paragraph of five sentences instead of five paragraphs of one sentence.

-Please provide more details about the GO and CNC or at least a link to the companies with a reference number to allow researchers in the future to repeat the experiment with materials with the same properties. For example, what was the original biomass used to produce the CNC? How was it manufactured? What is the aspect ratio of the crystals? These details may create enormous differences between the two experiments. 

2.2. Sample preparation:

-Please write this subsection in two paragraphs, not a numbered list.

-I strongly suggest presenting pictures of the nanocomposites with different G/C ratios. If the authors do not want to add them to the main manuscript to not cause problems in the flow, please add them to the Supplementary Materials.

2.3. Characterization tests

2.3.1. X-ray Photoelectron Spectra (XPS): OK

2.3.2. Raman spectra: OK

2.3.3. Thermogravimetric Analysis (TGA):

-The authors did not mention that they performed a TGA for the nanocomposites. Please add this information to allow the discussion in section 3.2.2.

2.3.4. Testing Mechanical Properties:

-Please explain how many replications were performed.

-I suggest measuring the work-to-break for the samples and calculating the areas under the curves.

2.3.5. Oxygen Transmission Rate (OTR):

-This paragraph looks more like an instruction manual than a procedure. Please rewrite this section as the other sections of the text. 

-Please explain how many replications were performed.

-What are the dimensions of the membranes? Is there a difference in the thickness based on the G/C ratio?

3. Results and discussion

3.1. Characterizations of Nanoparticles

3.1.1. XPS spectra

-I suggest adding the graphs of cellulose: graphene fraction x relative atomic percentages in Supplementary Materials, with the equations for the 2nd-order polynomial curves. The curves look like logistics, but I could not test them for a logistical regression, and the 2nd-order polynomial regression had a good fit.

3.1.2. Raman spectra:

-The authors claim the same thing twice with different words. The claim that decreasing G/C ratio increases the repair is the same as the claim that increasing G/C ratio decreases the repair. Please remove the last sentence.

3.1.3. Thermogravimetric analysis (TGA)

-The reasoning to test the thermal stability of the raw materials up to 700°C is unclear. All the reactions and tests happened at no more than 100°C, so why did the authors test these high temperatures? Is there any potential application for the nanocomposite that may occur above 100°C?

3.2. Properties of Nanocomposites

3.2.1. Mechanical Properties

-Figure 6 and Table 2 represent the same type of information. I recommend removing figure 6 and keeping only table 2 in the main document.

-I prepared graphs with the information from Table 2. I could not calculate a logistic regression with the data, only a 2nd order polynomial regression to show the authors that they can get more information from the data. The mechanical properties of the nanocomposites have a logistic trend in this dataset. Please make a logistic regression. If the information helps calculate the optimal ratio GC, please add the information to the main manuscript. If the authors consider exploring this optimum ratio in the future, please add it to the Supplementary Materials.

3.2.2. Thermal Stability

-The authors did not mention that they performed a TGA for the nanocomposites in section 2.3.3. Please add this information.

-Please perform a regression analysis. The data look, once again, as a logistic regression.

-Figure 8 and Table 3 represent the same type of information. Please keep Figure 8 in the main document and move Table 3 to the Supplementary Materials.

3.2.3. Gas Barrier Properties

-Figure 9 and Table 4 represent the same type of information. Please keep Figure 9 in the main document and move Table 4 to the Supplementary Materials.

-Please perform a regression analysis. The data look, once again, as a logistic regression.

4. Conclusions

-They are fine, but the authors could suggest studying other ratios in the future, such as 1:1 and the optimized ratios found in this study.

**Acknowledgments: OK

**CRediT:

-Please insert the participation of the authors based on the CRediT (Contribution Roles Taxonomy) - https://credit.niso.org/ 

**Conflict of interests:

-Please add the comment about possible conflicts of interest of the authors

**Data availability statement:

-Please add a data availability statement (https://www.mdpi.com/journal/data/instructions).

**References:

-Please avoid capitalizing all the nouns in the title of an article, even if it was published this way. Therefore, please write “Graphene-based membranes for molecular separation” instead of “Graphene-Based Membranes for Molecular Separation,” for example.

Round 2

Reviewer 3 Report (New Reviewer)

Dear Authors,

Thank you for considering comments and suggestion as well as for showing extra characterizations. 

Some issues persist:

1. Figures with very low resolution. The different ratios are not observed. 

2. The discussion of Figure 10 is unclear. What is the difference between Figure 1 and 10. How the Authors reach that illustration, say, Top and Bottom panels. 

3. Figure 1 from the point-by-point letter must be included in the main text. If it was used in a previous work, please provide discussion and related reference. If not, add and discuss it. I'm so curious why that result is avoided. 

4. The uncertainty is only added in some experiments. But the other ones? Why the uncertainty is not systematic?

5. Relying solely on the ID/IG ratio does not provide conclusive evidence for structural repair. The discussion appears to be ambiguous and lacks some references, suggesting that the authors may not wish to delve into a more in-depth examination of the topic.

Author Response

This manuscript is a resubmission of an earlier submission. The following is a list of the peer review reports and author responses from that submission.

Round 1

Reviewer 1 Report

The present paper deals with the synergistic use of two different nanoparticles in reinforcing mechanical and barrier properties, and this part of the results are very good and interesting. However, the interpretation of the results about a  defect restoration in GO nanosheets by cellulose nanocrystals is not acceptable in any case. To stand this hypothesis (I consider it impossible by simply mixing particles) other analytical techniques would be required. As a consequence, although there are valid results in terms of mechanical properties and barrier effect, my recommendation is that the present paper must be rejected.

The authors can reconsider other interpretation for their results rather than  to fix or repair GO sheets, and can consider to use other techniques such as TEM for nanoperticle dispersion in GC-PSF nanocomposites. Other comments follow:

GO defects: What are GO defects for the authors? Graphene oxide sheets present plenty of defects. Everything that is not a sp2C-sp2C honeycomb structure is a defect: sp3C atoms bonded any atom, functional groups, and of course edges, either sheet external or internal (holes). XPS defect contribution is due to which ones?

XPS interpretation: It is mostly wrong. Authors MUST read correct interpretation in C1s XPS range. Basically: 284.6 eV is due to sp2C-sp2C bond (delocalized), and in CNC there is no bonds of this type. sp3C-sp3C  bonds are in the range 285.0-285.4 eV (what authors consider defects) coud be this contribution, and CNC interpretation is wrong as main peak is at 285 eV. 286.3-286.7 eV is several types of sp3C-O bond (CNC has plenty of them, obviously). 287.5 eV is either O-sp3C-O or sp2C=O. and 288.9 eV is carboxyl. XPS plots (Fig 2) shows a merge of both GO and CNC, and authors have also to consider that there are infinite ways to fit XPS to voigt contributions. Holes cannot be monitored by XPS.

Raman interpretation: Similar comments apply. Raman basically get signals from sp2 carbons. So G signal comes from sp2 carbons in a pure graphitic vicinity, whereas D signal comes also from graphitic sp2 carbons but in a defect vicinity, and regularly are sharp peaks. D broad peaks is indicative of amorphous. As a consequence, there is no direct proportionality between ID/IG and defect except for pure graphitic/graphene, which is not the case for graphene oxide.

FTIR analysis would improve the characterization, as well as dispersion by TEM. This could help to know why there is a good synergy between GO and CNC for reinforcing greatly PSF.

Finally, there is a complete lack of details in the experimental section: section 2.2, the amoung of GO and CNC in 200 mL DMF; amount of PSF in 200 mL DMF. Is 0.5 wt% the amount of nanoparticles with respect to PSF? It is not clear. Raman laser wavelength is not reported. And more importantly, Graphene oxide is NOT the precursor of graphene.

Reviewer 2 Report

The authors should provide more details about how the Oxygen Transmission Rate was measured.

To prepare the PSF/GC composites (with a loading of GC of 0.5 wt%) the authors solubilized PSF in DMF and then added GC to the solution. In their discussion of PSF/GC composites the authors assume that the GC nanofillers (made of GO and CNC) remained intact (GO and CNC together) despite their dissolution in DMF (GO is soluble in DMF, CNC probably not). This assumption seems very weak (and probably wrong). During addition of GC to DMF, the GO and CNC may well separate and in the end the composite is of PSF+GO+CNC and not PSF+GC. The authors should address this major weakness.

On line 196 it says “This is indicative of the presence of hydrophilic interactions between CNC and GO, which enhance the thermal stability of GC nanoparticles.” – I do not see why this is indicative of hydrophilic interactions. This is pure speculation from the authors.

On line 199 it says: “In addition, relatively speaking, the thermal stability of GC 2:1 is the best, while the thermal stability of GC 32:1 is the worst, which also proves that G/C ratios have a significant impact on the interaction between GO and CNC.” – This is not true. This conclusion is not supported by the results. According to Figure 4, GC 32:1 is clearly more thermally stable than GC 2.1

On line 219 saying that “This can be attributed to the CNC repairing the structural defects on the surface of GO” seems pure speculation.

On line 256 saying that “the addition of CNCs to the nanocomposites plays a vital role in repairing the structural defects on the surface of GO, thereby preventing the direct penetration of gas molecules through the defects” seems pure speculation.

Quality of English is good enough

Reviewer 3 Report

Idea behind the manuscript “Investigation of Graphene Oxide with Structural Defects Re-2 paired by Cellulose Nanocrystals with Different Ratios in Poly-3 sulfone 4” can be appreciated, however the analysis techniques and results obtained have been characterized superficially and do not meet the standards of “Polymers”. Few suggestions for improvements:

1. The claim of graphene oxide with structural defects and repairing must be justified with more precise techniques like ‘Micro-Raman spectroscopy’, for better analysis and analyze impurity level.

2. Why HR-TEM studies missing for the neat and composited samples.

3. I think it is important to include the theoretical studies for better interpretation of these structures, and to publish in a high impact factor journal.

4. Revise introduction and conclusion section for focus and aim of study.

5. More recent relevant references can be included.

6. Grammar and typo check.

Please see above review report.
